# How to Train Your DRAGON:
# Diverse Augmentation Towards Generalizable Dense Retrieval

**Sheng-Chieh Lin[1\*], Akari Asai[2], Minghan Li[1], Barlas Oguz[3],**
**Jimmy Lin[1], Yashar Mehdad[3], Wen-tau Yih[3], and Xilun Chen[3†]**

University of Waterloo[1], University of Washington[2], Meta AI[3]

{s269lin,m692li,jimmylin}@uwaterloo.ca, akari@cs.washington.edu
{barlaso,mehdad,scottyih,xilun}@meta.com

## Abstract

Various techniques have been developed in recent years to improve dense retrieval (DR), such as unsupervised contrastive learning and pseudo-query generation. Existing DRs, however, often suffer from effectiveness tradeoffs between supervised and zero-shot retrieval, which some argue was due to the limited model capacity. We contradict this hypothesis and show that a generalizable DR can be trained to achieve high accuracy in both supervised and zero-shot retrieval without increasing model size. In particular, we systematically examine the contrastive learning of DRs, under the framework of Data Augmentation (DA). Our study shows that common DA practices such as query augmentation with generative models and pseudo-relevance label creation using a cross-encoder, are often inefficient and suboptimal. We hence propose a new DA approach with diverse queries and sources of supervision to progressively train a generalizable DR. As a result, DRAGON,[1] our **D**ense **R**etriever trained with diverse **AuG**mentati**ON**, is the first BERT-base-sized DR to achieve state-of-the-art effectiveness in both supervised and zero-shot evaluations and even competes with models using more complex late interaction.

## 1 Introduction

Bi-encoder based neural retrievers allow documents to be pre-computed independently of queries and stored, enabling end-to-end retrieval among huge corpus for downstream knowledge-intensive tasks (Karpukhin et al., 2020; Reimers and Gurevych, 2019). Recently, Thakur et al. (2021b) show that bi-encoder retrievers still underperform BM25 in real-world scenarios, where training data is scarce. One potential solution is

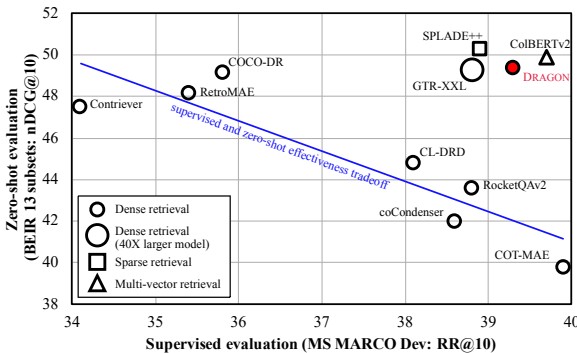

Figure 1: Supervised versus zero-shot effectiveness comparison among existing state-of-the-art retrievers. All models use a BERT-base-sized (110M parameters) backbone except for GTR-XXL (4.8B parameters).

to design more expressive representations to capture more fine-grained token-level information; e.g., SPLADE++ (Formal et al., 2022) and ColBERTv2 (Santhanam et al., 2022b) in Figure 1. However, these designs add complexity and latency to retrieval systems (Mackenzie et al., 2021).

By contrast, dense retrieval (DR) is a simpler bi-encoder retriever that maps queries and documents into low-dimensional vectors and computes text similarity through a simple dot product. Top-$k$ retrieval can be performed directly using ANN search libraries (Johnson et al., 2021; Guo et al., 2020). Recently, various methods have been proposed to improve DR effectiveness while keeping its simple architecture, such as pre-training (Lee et al., 2019; Chang et al., 2020), query augmentation (Oguz et al., 2022), and distillation (Ren et al., 2021; Zeng et al., 2022). For example, pre-training on MS MARCO corpus improves accuracy in the fully supervised setting while leveraging other corpora can improve transfer in the zero-shot setting. However, improvement in one setting is only achieved at the expense of the other. Figure 1 plots existing state-of-the-art DR models with respect to their effectiveness on these two axes, which presents a

---

*This work is done during Sheng-Chieh's internship at Meta.
†Xilun and Sheng-Chieh contributed equally to this work.

[1]The code and model checkpoints are available at: https://github.com/facebookresearch/dpr-scale

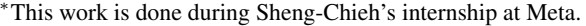

clear tradeoff between supervised and zero-shot effectiveness (the blue line). The only exception, GTR-XXL (Ni et al., 2022), breaks the effectiveness tradeoff at the expense of efficiency (i.e., query encoding), which leverages very large pre-trained models with 40 times more parameters. This effectiveness tradeoff prompts some to hypothesize that we have fully exploited the capacity of BERT-base-sized DR model (Ni et al., 2022) and explore how to cleverly increase model parameters without sacrificing retrieval efficiency. For example, the recent work (Wang et al., 2022; Dai et al., 2022) proposes to train one expert dense retriever for each specific scenario, resulting in slow adaptation to real-world applications (Asai et al., 2023).

In this work, we contradict this hypothesis and show that a generalizable DR can indeed be trained to achieve state-of-the-art effectiveness in both supervised and zero-shot evaluations *without* increasing model size. To this end, we first investigate the important factors contributing to the recent progress of DR. For example, DR seems to gain zero-shot transfer capability from pre-training on large-scale and diverse training queries (Izacard et al., 2021; Yu et al., 2022) while knowledge distillation can improve the supervision quality by automatically identifying relevant passages which are not labeled by humans (Ren et al., 2021; Zeng et al., 2022). To better understand these approaches, we devise a unified framework of data augmentation (DA) for contrastive learning. Under the framework, the previous work can be viewed as DA with different recipes of query augmentation and relevance label augmentation shown in Table 1. The DA framework also helps us design comprehensive studies for better DR training.

Guided by a detailed empirical exploration along the space of our DA framework, we find the following: (1) for relevance label augmentation, we identify that the key to training a generalizable dense retriever is to create *diverse* relevance labels for each query, for which we use multiple retrievers instead of a strong cross encoder; (2) with such diverse relevance labels, dense retrievers can be trained effectively using cheap and large-scale augmented queries (e.g., cropped sentences from a corpus) instead of the more expensive neural generative queries. This finding opens the door to further building cheap but useful training data in scale for DR in the future. Finally, we find that it is suboptimal for a dense retriever to learn the

Table 1: Categorization of existing DR models by their approaches to data augmentation.

| Model | Qry Aug. | Label Aug. | Corpus |
|---|---|---|---|
| RocketQAv2 (Ren et al., 2021) CL-DRD (Zeng et al., 2022) | ✗ | CE | MARCO |
| coCondenser (Gao and Callan, 2022) Contriever (Izacard et al., 2021) COCO-DR (Yu et al., 2022) | cropping | ✗ | MARCO Wiki+ CCnet BEIR |
| GPL (Wang et al., 2022) PTR (Dai et al., 2022) | GenQ | ✗ | BEIR |
| DRAGON | cropping+GenQ | retrievers | MARCO |

diverse relevance labels from multiple retrievers directly. Thus, we propose a simple strategy to *progressively* augment relevance labels which guides dense retrievers to learn diverse relevance signals more effectively.

Our final model is trained on 28 million augmented queries consisting of two types (cropped sentences and synthetic queries), as well as progressive relevance label augmentation using diverse (sparse, dense, and multi-vector) retrievers. As shown in Figure 1, DRAGON, a **D**ense **R**etriever trained with diverse **AuG**mentati**ON**, is the *first* dense retriever to break the supervised and zero-shot effectiveness tradeoff without increasing model size or retrieval complexity; e.g., GTR-XXL, SPLADE++ and ColBERTv2.

We summarize our contributions as follows: (1) We conduct a systematic study of DR training under the lens of data augmentation, which provides some surprising but key insights into training a generalizable dense retriever; (2) We propose a progressive label augmentation strategy to guide a dense retriever to learn the diverse but complex relevance labels; (3) DRAGON, our BERT-base-sized DR, reaches state-of-the-art retrieval effectiveness in both supervised and zero-shot evaluations.

## 2 Background

In this section, we first introduce the retrieval task and contrastive learning for dense retrieval. We then provide a unified framework for understanding recent approaches to improve dense retrieval training as instances of data augmentation.

### 2.1 Training Dense Retrieval Models

Given a query $q$, our task is to retrieve a list of documents to maximize some ranking metrics such as nDCG or MRR. Dense retrieval (DR) based on pre-trained transformers (Devlin et al., 2018; Raffel et al., 2020) encodes queries and documents as low dimensional vectors with a bi-encoder architecture and uses the dot product between the encoded

vectors as the similarity score:

$$s(q, d) \triangleq \mathbf{e}_{q_{\text{[CLS]}}} \cdot \mathbf{e}_{d_{\text{[CLS]}}}, \qquad (1)$$

where $\mathbf{e}_{q_{\text{[CLS]}}}$ and $\mathbf{e}_{d_{\text{[CLS]}}}$ are the [CLS] vectors at the last layer of BERT (Devlin et al., 2018).

Contrastive Learning is a commonly used method for training DR models by contrasting positive pairs against negatives. Specifically, given a query $q$ and its relevant document $d^+$, we minimize the InfoNCE loss:

$$-\log \frac{\exp(s(q, d^+))}{\exp(s(q, d^+)) + \sum\limits_{j=1}^{k} \exp(s(q, d_j^-))}. \quad (2)$$

## 2.2 A Unified Framework of Improved Dense Retrieval Training: Data Augmentation

Data augmentation (DA) for contrastive learning has been widely used in many machine learning tasks (Chen et al., 2020; Thakur et al., 2021a). In fact, many recent approaches to train better DR, such as knowledge distillation, contrastive pretraining and pseudo query generation (GenQ), can be considered DA with different recipes respectively listed in the first three main rows of Table 1. We compare the DA recipes from the perspectives of query and relevance label augmentation. We refer readers to Appendix A.9 for more related work of advanced DR training strategies.

**Query Augmentation.** There are two common automatic approaches to increase the size of training queries from a given corpus, sentence cropping and pseudo query generation. The former can easily scale up query size without any expensive computation, which is used by the models for contrastive pre-training (the second section of Table 1; Gao and Callan, 2022; Izacard et al., 2021; Wu et al., 2022). The latter generates quality but more expensive human-like queries using large language models for DR pre-training (Oguz et al., 2022) or domain adaptation (the third section of Table 1; Wang et al., 2022; Dai et al., 2022). Concurrently to our work, Meng et al. (2023) explore various approaches to query augmentation, such as span selection and document summarization.

**Relevance Label Augmentation.** The aforementioned approaches to query augmentation often assume that the (or part of the) original document is relevant to the augmented queries, which may not be true and only provides a single view of relevance labeling. The recent work (the first section of Table 1; Ren et al., 2021; Zeng et al., 2022) improve

DR training with the positive passages predicted by cross encoders. These knowledge distillation approaches further improve training data quality through label augmentation, inspiring us to conduct relevance label augmentation on the augmented queries (i.e., cropped sentences and GenQ).

## 2.3 Settings for Empirical Studies

We introduce some basic experimental settings to facilitate the presentation of our empirical studies on data augmentation in Section 3. More detailed settings can be found in Section 4. Following previous work (Izacard et al., 2021; Xiao et al., 2022; Yu et al., 2022; Formal et al., 2022; Santhanam et al., 2022b), we consider MS MARCO (Bajaj et al., 2016) as supervised data and BEIR datasets for zero-shot evaluations. Thus, we use the 8.8 million MS MARCO passage corpus to conduct data augmentation and evaluate our trained models on MS MARCO Dev, consisting of 6980 queries from the development set with one relevant passage per query on average. We report MRR@10 (abbreviated as RR@10) and Recall@1000 (R@1K) as the evaluation metrics. For zero-shot evaluations, we use BEIR (Thakur et al., 2021b), consisting of 18 IR datasets spanning diverse domains and tasks including retrieval, question answering, fact checking, question paraphrasing, and citation prediction. We report the averaged nDCG@10 over 13 public BEIR datasets, named BEIR-13, making the numbers comparable to most existing approaches (Formal et al., 2021; Santhanam et al., 2022b).[2]

## 3 Pilot Studies on Data Augmentation

In this section, we first discuss the exploration space of data augmentation (DA) based on the framework in Section 2.2 and then conduct empirical studies on how to better train a dense retriever. Based on the empirical studies, we propose our DA recipe to train DRAGON, a **D**ense **R**etriever with diverse **A**u**G**mentati**ON**.

### 3.1 An Exploration of Data Augmentation

**Query Augmentation.** Following the discussion in Section 2.2, we consider the two common approaches to automatic query augmentation. Specifically, for sentence cropping, following Chen et al. (2022), we use the collection of 28 million sentences from the MS MARCO corpus

---

[2]CQADupStack, Robust04, Signal-1M, TREC-NEWS, BioASQ are excluded

consisting of 8.8 million passages. As for pseudo query generation, we use the 28 million synthetic queries sampled from the query pool generated by doct5query (Nogueira and Lin, 2019). In addition, we also consider augmenting the type of queries by mixing cropped sentences and synthetic queries.

**Label Augmentation with Diverse Supervisions.** Although cross encoder (CE) is known to create relevance labels with strong supervision, we hypothesize that CE still cannot capture diverse matching signals between text pairs. A query is often relevant to many documents from different perspectives (e.g., semantic or lexical matching), which cannot capture by a single labeling scheme (a strong model or even human). In this work, we seek multiple sources of supervisions from existing sparse, dense and multi-vector retrievers, which are more efficient than CE and suitable for labeling a large number of queries (see discussion in Section 5).

## 3.2 Training with Diverse Supervisions

We have introduced our searching space for query and label augmentation (with diverse supervisions); however, training a dense retriever on such augmented data is not trivial. First, how can we create training data using a teacher from any augmented queries (i.e., cropped sentences or pseudo generative queries)? Second, with the training data sets created from multiple teachers, how can we train a dense retriever to digest the multiple supervisions?

Formally speaking, given $N$ teachers, for each augmented query $q$, we retrieve $N$ ranked lists (i.e., $\mathcal{P}_q^1, \mathcal{P}_q^2, \cdots, \mathcal{P}_q^N$ with each list has $K$ passages) from the corpus with the respective teachers. We consider the ranked list $\mathcal{P}_q^n$ from the $n$-th teacher a source of supervision since the top-$k$ and last-$k'$ passages in $\mathcal{P}_q^n$ contain the teacher's view on what is relevant and less relevant for the given query. We then discuss possible strategies to train a dense retriever with diverse supervisions.

**Fused Supervision.** An intuitive strategy is to fuse the multiple sources into a single high-quality supervision, which dense retrievers can learn from. For the augmented query $q$, we conduct linear score fusion (Ma et al., 2021) on the $N$ ranked lists to form a new ranked list $\mathcal{F}_q$ as a fused supervision.

**Uniform Supervision.** Another simple strategy is to provide a dense retriever with equal exposures to multiple sources of supervisions. Specifically, given a query, we uniformly sample a source of supervision; i.e., a ranked list $\mathcal{P}_q^n$, where $n \sim$

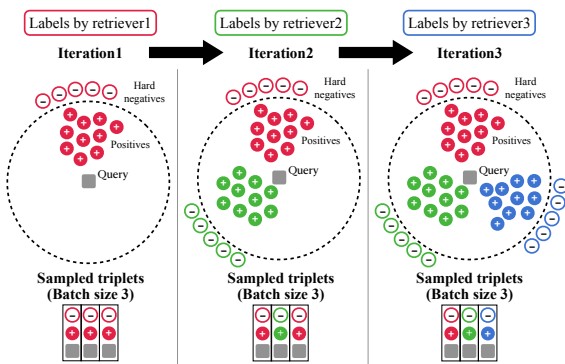

Figure 2: Illustration of progressive label augmentation. For each iteration of training, additional relevance labels from a teacher are augmented in the training data. By contrast, uniform supervision directly exposes models to all the supervisions (as in iteration 3) in the beginning.

$\mathcal{U}(1, N)$. This approach naturally encourages the positive samples appearing in more ranked lists to be sampled and vice versa. The advantage is that fusion weight tuning is not required. Furthermore, models can see diverse supervisions from different teachers in contrast to fused supervision, which may be dominated by a single strong teacher.

**Progressive Supervision.** The previous two approaches directly give models supervisions from multiple teachers at once; however, learning directly from the mixture of supervision is challenging, especially for DR models which compute text matching with simple dot product. Inspired by the success of curriculum learning (Zeng et al., 2022), we propose an approach to *progressive label augmentation* to guide DR training with progressively more challenging supervision. Specifically, we train our models with uniform supervision for $N$ iterations and at each iteration, we augment relevance label using additional teacher, as illustrated in Figure 2; i.e., at iteration $T \leq N$, we uniformly sample a source of supervision, $\mathcal{P}_q^n$, where $n \sim \mathcal{U}(1, T)$ (see Appendix A.6 for more study and explanation). A key factor of this approach is how to arrange the order for easy-to-hard supervisions; namely, the trajectory of progressive supervision.

With any aforementioned strategy to obtain diverse supervisions, we train our dense retrievers using contrastive loss in Eq. (2). Specifically, given a query $q$, we first obtain a source of supervision either from sampling ($\mathcal{P}_q^n$) or fusion ($\mathcal{F}_q$); then, we randomly sample a positive and hard negative from the top 10 passages and top 46–50 passages, respectively to form a triplet. The sampling scheme

Table 2: Strategies to obtain multiple supervisions using cropped sentences as queries.

| Teacher | 0 uniCOIL | 1 Contriever | 2 ColBERTv2 | 3 fused | 4 unif. | 5* prog. |
|---|---|---|---|---|---|---|
| | | | | three teachers | | |
| *effectiveness of student* | | | | | | |
| MARCO Dev | 34.9 | 33.9 | 36.4 | 36.7 | **36.9** | 36.6 |
| BEIR-13 | 46.7 | 47.0 | 46.3 | 46.6 | 47.7 | **49.3** |
| *effectiveness of teacher* | | | | | | |
| MARCO Dev | 35.1 | 34.1 | 39.7 | 40.0 | - | - |
| BEIR-13 | -△ | 47.5 | 49.9 | -△ | - | - |

\* The condition of column 5 corresponds to row 0 in Table 3.
△ We do not evaluate uniCOIL on BEIR due to its requirement of expensive document expansion from corpus.

Table 3: Study on trajectory of progressive supervision using cropped sentences as queries.

| Progressive supervision trajectories | MARCO dev RR@10 | BEIR-13 nDCG@10 |
|---|---|---|
| (0) uniCOIL→ Contriever → ColBERTv2 | 36.6 | **49.3** |
| (1) Contriever → uniCOIL→ ColBERTv2 | 36.7 | 48.4 |
| (2) ColBERTv2→ Contriever→ uniCOIL | 36.4 | 47.7 |
| (3) uniCOIL→ Contriever → ColBERTv2* | **36.8** | 47.4 |

\* ColBERTv2 is the only teacher at the last (3rd) iteration.

has been empirically proved to well preserve the supervised signal from a single teacher (Chen et al., 2022) (also see our study in Appendix A.5). In this work, we further extend the sampling scheme to obtain diverse supervisions from multiple teachers.

### 3.3 Empirical Studies

**Strategies to Obtain Diverse Supervisions.** We first conduct empirical studies on how to better train a dense retriever in a simplified setting by using the MS MARCO cropped sentences as augmented queries and obtain supervised labels using three teachers with diverse relevance score computation: uniCOIL (sparse), Contriever (dense) and ColBERTv2 (multi-vector). To compare the different strategies discussed in Section 3.2, We report the models trained with single (columns 0–2) and multiple (columns 3–5) sources of supervisions for 20 epochs and 60 epochs, respectively. For progressive supervision, we follow the supervision trajectory: uniCOIL→ Contriever → ColBERTv2 with 20 epochs for each of the three iterations ($N = 3$). Note that for fused supervision, we use MS MARCO Dev queries to tune and obtain the best hyperparameters to create fusion list.

The results are tabulated in Table 2. We observe that when learning from a single supervision (columns 0–2), there is a tradeoff between supervised and zero-shot retrieval effectiveness. Learning from the fusion list only sees a slight improvement over supervised evaluation while no improvement observes in zero-shot evaluations (columns 0–2 vs 3). By contrast, the model sees notable improvements in zero-shot evaluations when trained with uniform supervision (columns 0–3 vs 4), indi-

cating that learning from the diverse relevance labels from multiple retrievers separately rather than single strong supervision (ColBERTv2 or fused supervision) is key to gain generalization capability. Finally, we observe that progressive supervision can further guide a dense retriever to gain generalization capability over uniform supervision (column 4 vs 5). Thus, we use progressive supervision in the following experiments.

**Trajectory of Progressive Supervision.** We then study how to better arrange the trajectories of progressive supervision in Table 3. We observe that different trajectories have much impact on models' zero-shot retrieval effectiveness while a minor impact on supervised evaluation can be seen. For example, switching the sampling order between uniCOIL and Contriever results in a degrade of 1 point on the averaged nDCG@10 over BEIR-13 (row 0 vs 1) while reversing the whole trajectory leads to a degrade with more than 1.5 points (row 0 vs 3). This observation reflects an intuition that the retrievers with better generalization capability may capture more complex matching signal between text pairs (ColBERTv2 shows better generalization capability than the other two teachers); thus, their relevance labels should be augmented at a later stage of model training. Finally, in row 3, we follow the trajectory in row 0 but only use ColBERTv2 as the only source of supervision instead of obtaining uniform supervision from the three teachers at the last iteration. This change results in worse zero-shot retrieval effectiveness, indicating that learning from diverse supervisions is key to training a generalizable dense retriever.

**Query Augmentation.** We study the impacts of query augmentation on models' effectiveness under the scenario where supervision is given, which has not been studied so far in the field of dense retrieval. With the best trajectory of progressive supervision, we compare models trained using cropped sentences (rectangles), generative queries (GenQ; cir-

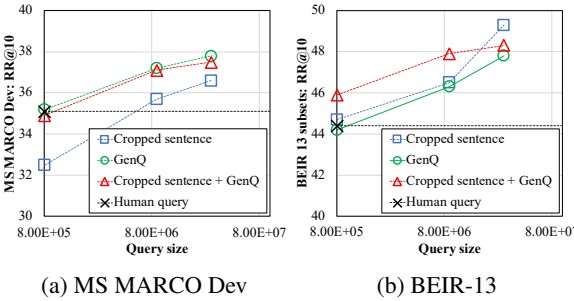

|                | (a) MS MARCO Dev | (b) BEIR-13 |
| --- | --- | --- |

Figure 3: Impacts of query augmentation.

cles), their mixture (triangles) and human queries (cross) in Figure 3. We observe that query size is the key to successful training. Although training with limited (0.8M) cropped sentences cannot perform well in MS MARCO dataset, scaling up the size (28M) sees significant improvement over the model trained on 0.8M human queries. Similarly, in Figure 3 (b), model's generalization capability shows a huge jump when scaling up query size. While surprisingly, cropped sentences can help dense retrievers to gain more generalization capability than human-like GenQ, a mixture of cropped sentences and GenQ yields strong retrieval effectiveness in supervised and zero-shot evaluations, when the query size is not large enough (0.8–8M).

## 3.4 Training our DRAGONs

With the empirical studies on DR training, we then propose the final recipe to train our DRAGON. We train DRAGON for 20 epochs (around 130K steps) at each iteration, with the trajectory of progressive supervision: uniCOIL→ Contriever → GTR-XXL → ColBERTv2 → SPLADE++. We list all the teacher model checkpoints for label augmentation in Appendix A.2. This trajectory is based on models' retrieval effectiveness on BEIR with the intuition gained from Section 3.3 that a more generalizable model creates relevance labels with more complex matching signals. For query augmentation, we mix half of cropped sentences and synthetic queries as training queries. Note that we do not further fine-tune our models on the MS MARCO training queries. In addition, we train other three DRAGON variants. DRAGON-S and DRAGON-Q only use cropped sentences and synthetic queries, respectively. As for DRAGON+, we follow the same training procedure of DRAGON but switch the initialization from BERT to the masked auto-encoding pre-trained model, RetroMAE.[3] We

[3] https://huggingface.co/Shitao/RetroMAE

will discuss the impacts of initialization in Section 5. The implementation of DRAGONs and the fully augmented training data are detailed in Appendix A.3 and A.4, respectively.

## 4 Comparison with the State of the Art

### 4.1 Datasets and Baseline Models

In addition to MS MARCO development queries, we evaluate model supervised effectiveness on the TREC DL (Craswell et al., 2019, 2020) queries, created by the organizers of the 2019 (2020) Deep Learning Tracks at the Text REtrieval Conferences (TRECs), where 43 (53) queries with on average 95 (68) graded relevance labels per query (in contrast to 6980 queries with on average 1 non-graded relevance label per query in MS MARCO Dev) are released. We report nDCG@10, used by the organizers as the main metric. For zero-shot evaluations, we report models' effectiveness on all the 18 datasets in BEIR (Thakur et al., 2021b). In addition, we use LoTTE (Santhanam et al., 2022b) consisting of questions and answers posted on StackExchange with five topics including writing, recreation, science, technology, and lifestyle. We evaluate models' retrieval effectiveness in the pooled setting, where the passages and queries from the five topics are aggregated. Following Santhanam et al. (2022b), the retrieval effectiveness of Success@5 on search and forum queries are reported. The detailed evaluation on LoTTE is listed in Appendix A.8.

We compare DRAGONs with dense retrievers using the backbone of bert-base-uncased trained with advanced techniques, such as knowledge distillation (Ren et al., 2021; Zeng et al., 2022), contrastive pre-training (Gao and Callan, 2022; Izacard et al., 2021; Yu et al., 2022), masked auto-encoding pre-training (Wu et al., 2022; Xiao et al., 2022) and domain adaptation (Wang et al., 2022; Dai et al., 2022). We refer readers to more detailed baseline model descriptions in Appendix A.1.

### 4.2 Results

**Supervised Evaluations.** The first main row in Table 4 reports models' retrieval effectiveness on MS MARCO passage ranking dataset. We first observe that some baseline dense retrievers which perform well in MS MARCO Dev set are either pre-trained on MS MARCO corpus (coCondenser and COT-MAE) or well fine-tuned on MS MARCO training queries with cross-encoder distillation (CL-DRD

Table 4: Comparison with existing state-of-the-art dense retrievers. Bold (underline) denotes the best (second best) effectiveness for each row among baseline dense models.

| Rep type | sparse | mul-vec | dense | baseline dense | | | | | | | | | | our dense | | | |
|---|---|---|---|---|---|---|---|---|---|---|---|---|---|---|---|---|---|
| | 0 | 1 | 2 | 3 | 4 | 5 | 6 | 7 | 8 | 9 | A | B | C | D | E | F |
| | SPLADE++ | ColBERTv2 | GTR-XXL | CL-DRD | RocketQAv2 | COT-MAE | RetroMAE | coCondenser | Contriever | COCO-DR | GPL | PTR | DRAGON-S | DRAGON-Q | DRAGON | DRAGON+ |
| Pre-training | ✓ | ✗ | ✓ | ✗ | ✗ | ✓ | ✓ | ✓ | ✓ | ✓ | ✗ | ✓ | ✗ | ✗ | ✗ | ✓ |
| Distillation | ✓ | ✓ | ✓ | ✓ | ✓ | ✗ | ✗ | ✗ | ✗ | ✗ | ✓ | ✗ | ✓ | ✓ | ✓ | ✓ |
| Target Corpus† | ✗ | ✗ | ✗ | ✗ | ✗ | ✗ | ✗ | ✗ | ✓ | ✓ | ✓ | ✓ | ✗ | ✗ | ✗ | ✗ |
| | | | | | | | | **MS MARCO (Supervised)** | | | | | | | | |
| Dev (RR@10) | 38.9 | 39.7 | 38.8 | 38.1 | 38.8* | **39.9*** | 35.4 | 38.6* | 34.1 | 35.8 | - | - | 38.1 | 39.1 | 39.3 | 39.0 |
| Dev (R@1K) | 98.2 | 98.4 | 99.0 | 97.9 | 98.1* | 98.5* | 97.5 | 98.4* | 97.9 | 97.9 | - | - | 98.3 | **98.8** | 98.5 | 98.6 |
| DL2019 (nDCG@10) | 74.3 | 74.6 | - | 72.5 | - | 70.0* | 68.8 | 71.5* | 67.8 | 74.1 | - | - | 73.6 | 74.0 | 74.1 | **74.4** |
| DL2020 (nDCG@10) | 71.8 | 75.2 | - | 68.3 | - | 67.8* | 71.4 | 68.1* | 66.1 | 69.7 | - | - | 70.0 | 72.6 | **72.9** | 72.3 |
| nDCG@10 | | | | | | | | **BEIR (Zero-shot)** | | | | | | | | |
| TREC-COVID | 71.1 | 73.8 | 50.1 | 58.4 | 67.5 | 56.1 | 77.2 | 71.2 | 59.6 | **78.9** | 70.0 | 72.7 | 73.9 | 73.2 | 74.0 | 75.9 |
| NFCorpus | 34.5 | 33.8 | 34.2 | 31.5 | 29.3 | 32.1 | 30.8 | 32.5 | 32.8 | **35.5** | 34.5 | 33.4 | 32.2 | 33.0 | 32.9 | 33.9 |
| FiQA-2018 | 35.1 | 35.6 | 46.7 | 30.8 | 30.2 | 28.3 | 31.6 | 27.6 | 32.9 | 31.7 | 34.4 | **40.4** | 35.6 | 35.3 | 35.0 | 35.6 |
| ArguAna | 52.1 | 46.3 | 54.0 | 41.3 | 45.1 | 27.8 | 43.3 | 29.9 | 44.6 | 49.3 | **55.7** | 53.8 | 51.5 | 45.5 | 48.9 | 46.9 |
| Tóuche-2020 | 24.4 | 26.3 | 25.6 | 20.3 | 24.7 | 21.9 | 23.7 | 19.1 | 23.0 | 23.8 | 25.5 | **26.6** | 26.5 | 26.0 | 24.9 | 26.3 |
| Quora | 81.4 | 85.2 | 89.2 | 82.6 | 74.9 | 75.6 | 84.7 | 85.6 | 86.5 | 86.7 | 83.6 | - | 86.4 | 87.1 | 86.9 | **87.5** |
| SCIDOCS | 15.9 | 15.4 | 16.1 | 14.6 | 13.1 | 13.2 | 15.0 | 13.7 | 16.5 | 16.0 | **16.9** | 16.3 | 15.9 | 15.0 | 15.4 | 15.9 |
| SciFact | 69.9 | 69.3 | 66.2 | 62.1 | 56.8 | 60.1 | 65.3 | 61.5 | 67.7 | **70.9** | 67.4 | 62.3 | 67.8 | 67.2 | 67.5 | 67.9 |
| NQ | 54.4 | 56.2 | 56.8 | 50.0 | 50.5 | 48.3 | 51.8 | 48.7 | 49.5 | 50.5 | 48.3 | - | 53.3 | 52.3 | 53.1 | **53.7** |
| HotpotQA | 68.6 | 66.7 | 59.9 | 58.9 | 53.3 | 53.6 | 63.5 | 56.3 | 63.8 | 61.6 | 58.2 | 60.4 | 65.6 | 62.7 | 64.8 | **66.2** |
| DBPedia | 44.2 | 44.6 | 40.8 | 38.1 | 35.6 | 35.7 | 39.0 | 36.3 | 41.3 | 39.1 | 38.4 | 36.4 | 40.6 | 41.4 | 41.4 | **41.7** |
| FEVER | 79.6 | 78.5 | 74.0 | 73.4 | 67.6 | 50.6 | 77.4 | 49.5 | 75.8 | 75.1 | 75.9 | 76.2 | 76.4 | 75.1 | 75.8 | **78.1** |
| Climate-FEVER | 22.8 | 17.6 | 26.7 | 20.4 | 18.0 | 14.0 | 23.2 | 14.4 | **23.7** | 21.1 | 23.5 | 21.4 | 21.8 | 20.3 | 22.2 | 22.7 |
| CQADupStack | 34.1 | - | 39.9 | 32.5 | - | 29.7 | 34.7 | 32.0 | 34.5 | **37.0** | 35.7 | - | 35.9 | 34.4 | 35.2 | 35.4 |
| Robust04 | 45.8 | - | 50.6 | 37.7 | - | 30.8 | 44.7 | 35.4 | 47.6 | 44.3 | 43.7 | - | 46.3 | 45.3 | 47.2 | **47.9** |
| Signal-1M | 29.6 | - | 27.3 | 28.2 | - | 21.1 | 26.5 | 28.1 | 19.9 | 27.1 | 27.6 | - | 29.3 | 28.9 | **30.1** | 30.1 |
| TREC-NEWS | 39.4 | - | 34.6 | 38.0 | - | 26.1 | 42.8 | 33.7 | 42.8 | 40.3 | 42.1 | - | 42.1 | 40.0 | 41.6 | **44.4** |
| BioASQ | 50.4 | - | 32.4 | 37.4 | - | 26.2 | 42.1 | 25.7 | 38.3 | 42.9 | **44.2** | - | 41.3 | 41.9 | 41.7 | 43.3 |
| Averaged nDCG@10 | | | | | | | | | | | | | | | | |
| PTR 11 subsets | 47.1 | 46.2 | 44.9 | 40.9 | 40.1 | 35.7 | 44.5 | 37.4 | 43.8 | 45.7 | 45.5 | 45.5 | 46.2 | 45.0 | 45.7 | **46.5** |
| BEIR-13 | 50.3 | 49.9 | 49.3 | 44.8 | 43.6 | 39.8 | 48.2 | 42.0 | 47.5 | 49.2 | 48.6 | - | 49.8 | 48.8 | 49.4 | **50.2** |
| Avg. all 18 datasets | 47.4 | - | 45.8 | 42.0 | - | 36.2 | 45.4 | 38.9 | 44.5 | 46.2 | 45.9 | - | 46.8 | 45.8 | 46.6 | **47.4** |
| Success@5 | | | | | | | | **LoTTE (Zero-shot)** | | | | | | | | |
| Search (pooled) | 70.9 | 71.6 | - | 65.8 | 69.8 | 63.4 | 66.8 | 62.5 | 66.1 | 67.5 | - | - | 71.4 | 72.4 | 72.6 | **73.5** |
| Forum (pooled) | 62.3 | 63.4 | - | 55.0 | 57.7 | 51.9 | 58.5 | 52.1 | 58.9 | 56.8 | - | - | 61.1 | 61.2 | 61.4 | **62.1** |

† The approach assumes the target corpus (e.g., BEIR) is available while training.

* These numbers are not comparable due to the use of non-standard MS MARCO corpus augmented with title (Lassance and Clinchant, 2023).

and RocketQAv2). However, their retrieval effectiveness on MS MARCO Dev set is not well correlated to TREC DL queries, which have fine-grained human labels with different degrees of relevance. We hypothesize that these models are able to retrieve the most relevant passage from the corpus but cannot retrieve diverse passages with different degrees of relevance. By contrast, all the variants of DRAGON trained with diverse augmented relevance labels show consistently strong effectiveness in MS MARCO Dev and TREC DL queries.

**Zero-Shot Evaluations.** The second main row in Table 4 reports models' zero-shot retrieval effectiveness on the BEIR datasets. We observe a reverse trend that those dense retrievers performing relatively poorly in MS MARCO Dev queries have

better zero-shot retrieval effectiveness (e.g., Contriever, COCO-DR and RetroMAE). These models are pre-trained (with data augmentation) on a corpus other than MS MARCO to combat domain shift issue in dense retrieval (Xin et al., 2022; Yu et al., 2022). On the other hand, DRAGONs trained on augmented data from MS MARCO corpus only transfer well to BEIR datasets. Furthermore, DRAGON+ reaches state-of-the-art retrieval effectiveness on BEIR as the sparse retriever, SPLADE++.[4] In addition, all the DRAGON variants outperform other dense retrievers by a large margin and compete SPLADE++ and ColBERTv2 in the LoTTE dataset. It is worth mentioning that the models trained with domain adaptation (columns

[4]See our BEIR leaderboard submission here.

Table 5: Label augmentation with cross encoder (CE) using cropped sentences as queries.

| | 0[†] | 1 | 2 | 3 |
|---|---|---|---|---|
| DRAGON-S initialization | | ✓ | ✓ | |
| Source of supervision | | all*+ CE | CE only | CE only |
| MARCO Dev (RR@10) | **38.1** | **38.1** | 37.5 | 36.8 |
| BEIR-13 (nDCG@10) | **49.8** | 49.7 | 48.7 | 47.7 |

[†] Column 0 corresponds to DRAGON-S.
* "all" denotes the five teachers used for training DRAGON-S.

Table 6: Ablation on initialized checkpoint using the mixture of cropped sentences and GenQ as queries.

| Initialized checkpoint | MARCO dev | BEIR-13 |
|---|---|---|
| | RR@10 | nDCG@10 |
| (0) BERT base (DRAGON) | 39.3 | 49.4 |
| (1) Contriever | 38.7 | 49.3 |
| (2) RoBERTa base | **39.4** | 49.9 |
| (3) RetroMAE (DRAGON+) | 39.0 | **50.2** |

9–B) perform better than the others (columns 3–8) but still underperform DRAGON in zero-shot evaluations. Using DRAGON as the base model for domain adaptation is possible to gain DR zero-shot effectiveness, which we leave to our future work.

A comparison between DRAGON-S and DRAGON-Q (columns C and D) shows that augmented query type has an impact on retrieval effectiveness in different datasets. DRAGON-S trained on cropped sentences surprisingly sees the highest retrieval effectiveness on BEIR but only sacrifices a bit on MS MARCO datasets. This means that cropped sentences, the cheap query type (compared to neural generative queries), are sufficiently helpful for models to learn domain-invariant retrieval capability. By contrast, we observe that DRAGON-Q trained with human-like queries performs poorly compared to DRAGON-S on the datasets where queries are far different from human-like queries, such as ArguAna (45.5 vs 51.5) and CQADupStack (34.4 vs 35.9), while mixing different types of queries (DRAGON) can mitigate the issue. Finally, DRAGON+, combined with masking auto-encoding (MAE) pre-training and our approach, sees further improvement on zero-shot evaluations without sacrificing in-domain ones, indicating that MAE pre-training may be orthogonal to our approach based on contrastive learning.

To sum up, DRAGONs advance state-of-the-art zero-shot effectiveness while keeping strong effectiveness in supervised evaluation. The experimental results demonstrate that our data augmentation approaches enable dense retrievers to learn domain-invariant matching signal between text pairs as the other models with fine-grained late interaction (SPLADE++ and ColBERTv2) or 40 times larger model size (GTR-XXL).

## 5 Discussions

**Is it necessary to augment relevance labels with a cross encoder?** To answer this question, we further train DRAGON-S with the augmented relevance labels from a cross encoder (CE). Specifically, we create a ranked list of CE by first retrieving top 1000 passages with DRAGON-S and re-ranking them with the CE for each cropped sentence as a query. With the CE ranked list, we conduct another iteration (20 epochs) of training for DRAGON-S; however, we do not see retrieval effectiveness improvement in Table 5 (column 0 vs 1). In addition, the retrieval effectiveness becomes even worse when we further train DRAGON-S by only sampling CE ranked list instead of uniformly sampling all the six ranked lists (column 1 vs 2). Finally, we initialize from bert-base-uncased and re-train the model for three iterations (60 epochs) only with the CE ranked list.[5] We observe that its effectiveness (column 3) is even worse than the models trained with the ranked lists from three retrievers (see columns 4 and 5 in Table 2). This result contradicts the general belief that CE provides the strongest supervision to a dense retriever (Hofstätter et al., 2020). Moreover, it demonstrates the effectiveness of using diverse supervisions to train a generalizable dense retriever, rather than relying on a single strong supervision. Furthermore, leveraging all the retrievers to augment labels is still more efficient than a cross encoder (see Appendix A.7).

**Does DRAGON benefit from unsupervised pre-training?** Table 6 compares the models trained from different checkpoint initialization. Note that for Contriever and RetroMAE, we initialize from the checkpoint with only unsupervised pre-training (without fine-tuning on MS MARCO). We observe that our approach benefits from further masked auto-encoding rather than contrastive pre-training (rows 2,3 vs 1). The result is sensible since our

---

[5]We do not notice effective improvement with more iterations of training both in supervised and zero-shot evaluations.

| Passage: The Manhattan Project and its atomic bomb helped bring an end to World War II. Its legacy of peaceful uses of atomic energy continues to have an impact on history and science. | | | | |
|---|---|---|---|---|
| Query Augmentation | | | Augmented rel (#) | Example of augmented rel |
| Cropping | Q1 | The **Manhattan Project** and its **atomic bomb** helped bring an end to World War II | 30 | The **Manhattan project** was a secret research and development project of the U.S to develop the atomic bomb. Its success granted the U.S the bombs that ended the war with Japan as well as ushering the country into the atomic era. |
| | Q2 | Its legacy of peaceful uses of **atomic energy** continues to have an impact on history and science. | 30 | An early nuclear power plant that used **atomic energy** to generate electricity. The Atomic Age, also known as the Atomic Era, is the period of history following the detonation of the first nuclear (atomic) bomb, ... |
| GenQ | Q1 | what were a major contributions to the **manhattan** effort | 26 | The **Manhattan Project** was an effort during World War II in the United States to develop the first nuclear weapon. |
| | Q2 | what impact did the **manhattan project** have on history | 26 | The **Manhattan Project**, which included some of history's greatest scientific minds, lead to the end of the war against the Japanese. But was it worth the environmental and financial costs? This massive site provides loads of ... |

Figure 4: Examples of augmented queries and relevance labels from a passage. Augmented rel (#) denotes the number of unique relevant passages labeled by all our five teachers.

approach can be considered an improved version of contrastive pre-training, which appears to be orthogonal to masked auto-encoding pre-training. We leave the investigation of improving DR with generative and contrastive pre-training combined for future research.

**Can we use the soft labels from multiple teachers?** In the literature, using the relevance scores from a teacher as soft labels is a standard of knowledge distillation (Lin et al., 2021b; Hofstätter et al., 2021). However, in our study, even when training with uniform supervision from a sparse and dense retriever (i.e., uniCOIL and Contriever), it is challenging to normalize their relevance scores and create universal soft labels, yielding significant supervised and zero-shot effectiveness drops. We suspect that dense and sparse retrievers have many different views on relevance score computation; thus, it is even harder for a dense retriever to learn the score distributions from the different teachers.

**Why sentence cropping yields a generalizable dense retriever?** Figure 4 showcases the augmented queries by sentence cropping and neural generation and their respectively augmented relevant passages other than the original passages. We observe two main differences between the cropped sentences and generative queries. Cropped sentences provide diverse queries from a passage; i.e., the two cropped sentences in Figure 4 include slightly different topics (Manhattan Project and atomic energy). By contrast, all generative queries surround the same main topic, Manhattan Project, about the original passages. Second, the cropped sentences have more unique augmented relevant passages than generative queries. This is maybe because a cropped sentence, containing more information (keywords), is more challenging than a generative human-like query. Thus, teachers show more disagreement between each other on cropped sentences. We hypothesize that a dense retriever trained on cropped sentences can capture more diverse supervised signals from multiple teachers than generative queries. This explains the reason why DRAGON-S shows better generalization capability than DRAGON-Q.

## 6 Conclusion

We present DRAGON, a **D**ense **R**etriever trained with diverse **AuG**mentati**ON** and a unified framework of data augmentation (DA) to understand the recent progress of training dense retrievers. Based on the framework, we extensively study how to improve dense retrieval training through query and relevance label augmentation. Our experiments uncover some insights into training a dense retriever, which contradicts common wisdom that cross encoder is the most effective teacher and human-like queries are the most suitable training data for dense retrieval. Then, we propose a diverse data augmentation recipe, query augmentation with the mixture of sentence cropping and generative queries, and progressive relevance label augmentation with multiple teachers.

With our proposed recipe of DA, DRAGON is the first to demonstrate that a single BERT-base-sized dense retriever can achieve state-of-the-art effectiveness in both supervised and zero-shot retrieval tasks. We believe that DRAGON can serve as a strong foundation retrieval model for domain adaptation retrieval tasks (Wang et al., 2022; Dai et al., 2022) or the existing retrieval augmented language models (Izacard et al., 2022; Shi et al., 2023; Mallen et al., 2023).

## Limitations

Despite of the easy usage of single-vector dense retrieval compared to the models with more fine-grained late interactions (e.g., SPLADE++ and ColBERTv2), the limitations of DRAGONs are mainly from the cost of training. First, to conduct diverse relevance label augmentation, well trained dense, sparse and multi-vector retrievers are required. Second, to optimize DRAGONs' effectiveness, we scale up training queries to the size of 28 millions (compared to 0.8 millions in MS MARCO training queries) and leverage the progressive training strategy, which costs five days of training time with 32 A100 (40 GB) GPUs. The training cost can be reduced by removing repetitive or meaningless queries, which we leave for future work.

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

## A Appendices

### A.1 Baseline Models

We compare DRAGON with dense retrievers using the backbone of bert-base-uncased trained with advanced techniques. (1) Knowledge Distillation: RocketQAv2 (Ren et al., 2021) distills knowledge from a cross encoder while CL-DRD (Ren et al., 2021) combines curriculum learning and cross-encoder distillation. They all use cross encoders' knowledge to augment positive relevance labels as our approach. (2) Contrastive Pre-training: coCondenser (Gao and Callan, 2022), Contriever (Izacard et al., 2021) and COCO-DR (Yu et al., 2022) are first pre-trained on different corpus listed in Table 1, and then fine-tuned on MS MARCO training queries. (3) Masked Auto-Encoding Pre-Training: COT-MAE (Wu et al., 2022) and RetroMAE (Xiao et al., 2022) are first pre-trained to recover polluted sentences and then fine-tuned on MS MARCO training queries. For RetroMAE, we use the variant with the best BEIR retrieval effectiveness for comparison. (4) Domain adaptation: We consider GPL (Wang et al., 2022) and Promptagator (PTR; Dai et al., 2022), which use generative models to create pseudo relevance data for each corpus in BEIR and train one expert dense retriever for each corpus. This approach requires the target corpus while training. Note that COCO-DR can also be considered domain adaptation on BEIR although it uses one model for all tasks.

Note that coCondenser and COT-MAE are fine-tuned on the "non-standard" MS MARCO passage corpus that has been augmented with title. Thus, we also conduct inference on the corpus with title for them; otherwise, we use the official MS MARCO passage corpus. In addition, we also report the retrieval effectiveness of GTR-XXL (Ni et al., 2022) and ColBERTv2 (Santhanam et al., 2022b) from their original papers and conduct retrieval for SPLADE++ using Pyserini (Lin et al., 2021a) for reference. We list all the other model checkpoints used for evaluations in Appendix A.2.

### A.2 Model Checkpoints

**Teacher Models:** (1) uniCOIL: `https://hu ggingface.co/castorini/unicoil-m smarco-passage`; (2) Contriever: `https: //huggingface.co/facebook/contri ever-msmarco`; (3) GTR-XXL: `https://hu ggingface.co/sentence-transformer s/gtr-t5-xxl`; (4) ColBERTv2: `https://`

`github.com/stanford-futuredata/Co lBERT`; (5) SPLADE++: `http://download -de.europe.naverlabs.com/Splade_ Release_Jan22/splade_distil_CoCo denser_medium.tar.gz`; (6) Cross encoder: `https://huggingface.co/cross-enc oder/ms-marco-MiniLM-L-12-v2`.

**Baseline Models:** (1) CL-DRD: `https://gi thub.com/HansiZeng/CL-DRD`; (2) Rock-etQAv2: we directly copy the numbers from San-thanam et al. (2022b); (3) COT-MAE: `https: //huggingface.co/caskcsg/cotma e_base_msmarco_retriever`; (4) Retro-MAE: `https://huggingface.co/Shita o/RetroMAE_BEIR`; (5) coCondenser: `https: //huggingface.co/Luyu/co-conde nser-marco-retriever`; (6) Contriever: `https://huggingface.co/faceboo k/contriever-msmarco`; (7) COCODR: `https://huggingface.co/OpenMatch /cocodr-base-msmarco`; (8) Promptagator (PTR) and GPL: we directly copy the numbers from their original papers (Dai et al., 2022; Wang et al., 2022).

### A.3 Implementation Details

We train our dense retrievers initialized from bert-base-uncased on 32 A100 GPUs (40GB) with a per-GPU batch size of 64 and a learning rate of $3e-5$. Each batch includes an augmented query with its positives and hard negatives. Following Karpukhin et al. (2020), we use asymmetric dual encoder with two distinctly parameterized encoders and leverage in-batch negative mining. Note that symmetric dual encoder shows poor generalization capability in our initial experiments. We set the maximum query and passage lengths to 32 and 128 for MS MARCO training and evaluation. For BEIR evaluation, we set maximum input lengths to 512.

Table 7: MS MARCO and our augmented training queries statistics.

|  | number | Avg. # tokens | Avg. # rel |
|---|---|---|---|
| passages in corpus | 8,841,823 | 78.8 | na |
| training queries | 532,761 | 8.2 | 1.0 |
| **augmented training queries** | | | |
| cropped sentences | 28,545,938 | 24.4 | 23.1 |
| generative queries | 28,545,938 | 8.0 | 24.7 |
| **test queries** | | | |
| Dev | 6,980 | 7.8 | 1.1 |
| DL2019 | 43 | 7.6 | 95.4 |
| DL2020 | 54 | 7.5 | 68.0 |

## A.4 MS MARCO Dataset Statistics

Table 7 lists the data statistics of MS MARCO dataset, including the original training queries and test queries (i.e., Dev, DL19 and DL20). In addition, we also list the augmented queries used to train DRAGONs with full relevance label augmentation by five teachers.

## A.5 Impacts of Top-$k$ Positive Sampling

Table 8: Ablation on progressive label augmentation from top-$k$ passages using cropped sentences as queries.

| top-$k$ positives | 1 | 5 | 10 |
|---|---|---|---|
| MARCO Dev (RR@10) | 33.1 | 36.4 | 36.6 |
| BEIR-13 (nDCG@10) | 42.4 | 48.0 | 49.3 |

* Trajectory: uniCOIL → Contriever → ColBERTv2.

In Section 3.2, we mention that our sampling scheme treats top 10 passages from each teacher ranked list as positives and top 45–50 as negatives. We further conduct experiment to study the impact of the positive sampling scheme. Following the experiment setups in Section 3.3, we use the sentences cropped from MS MARCO corpus as augmented queries and we conduct progressive label augmentation using top-$k$ passages as positive. The results are tabulated in Table 8. We observe that treating top-10 passages from each teacher as positives yields the best supervised and zero-shot effectiveness. On the other hand, using only the top passage as positive results in significant effectiveness drop. This result indicates that the top passage labeled by a teacher cannot transfer its knowledge well to a student. This result is similar to the observation from Chen et al. (2022).

## A.6 An Intuition Behind Uniform and Progressive Supervisions

As shown in Section 3.2, uniform supervision provides good supervision without fusion weight tuning as fused supervision. Intuitively, a positive retrieved by more teachers has a higher probability to be sampled and may be more relevant to a query. To provide a sense of why uniform supervision works, we estimate the accuracy of supervision by computing the probability of each positive sampled under uniform supervision, and rank the positives according to the simulated probability. For example, at the 3rd iteration of progressive training, given a query, a positive passage is labeled positive by all the three teachers, the probability of the passages being sampled is $\frac{1}{3} \cdot (\frac{1}{k} + \frac{1}{k} + \frac{1}{k}) = \frac{1}{k}$. In our experiments,

Table 9: Uniform and progressive supervision effectiveness comparison at each training iteration. The models are trained using cropped sentences as queries.

| Teacher / iteration | uniform | | | progressive | | |
|---|---|---|---|---|---|---|
| | $1 \rightarrow$ | $2 \rightarrow$ | 3 | $1 \rightarrow$ | $2 \rightarrow$ | 3 |
| uniCOIL | ✓ | ✓ | ✓ | ✓ | ✓ | ✓ |
| Contriever | ✓ | ✓ | ✓ | ✗ | ✓ | ✓ |
| ColBERTv2 | ✓ | ✓ | ✓ | ✗ | ✗ | ✓ |
| MARCO Dev | 36.2 | 37.0 | 36.9 | 34.9 | 35.8 | 36.6 |
| BEIR-13 | 46.6 | 47.4 | 47.6 | 46.7 | 48.6 | 49.3 |
| | effectiveness of teacher | | | | | |
| MARCO Dev | 39.1 | 39.1 | 39.1 | 35.1 | 36.5 | 39.1 |
| | diversity of teacher | | | | | |
| Avg. # rel | 17.5 | 17.5 | 17.5 | 10.0 | 14.9 | 17.5 |

each teacher labels the top 10 ($k = 10$) retrieved passages as positives in our labeling scheme. Note that, in the case where multiple positives have equal probability, we further rank them according to their sum of reciprocal rank. For instance, if the two passages (e.g., $p_1$ and $p_2$) are retrieved by all the three teachers; then, we further rank them according to their scores $\frac{1}{r_{11}} + \frac{1}{r_{12}} + \frac{1}{r_{13}}$ and $\frac{1}{r_{21}} + \frac{1}{r_{22}} + \frac{1}{r_{23}}$, where $r_{mn}$ denotes the rank of the passage $p_m$ by the $n$-th teacher. In addition, we also estimate the diversity of supervision by computing the number of positive passages in union sets from the sources of supervisions.

Table 9 reports the detailed effectiveness and supervision quality (accuracy and diversity) comparison at each training iteration between uniform and progressive supervision as discussed in our pilot study. We observe that uniform supervision provides accurate and diverse supervision in the beginning of training; however, the generalization improvement over iteration is less than progressive supervision.

## A.7 Latency Measurement for Relevance Label Augmentation

We measure the latency of label augmentation using batch retrieval on a single NVIDIA A100 40GB GPU for GPU search and 60 Intel(R) Xeon(R) Platinum 8275CL CPUs @ 3.00GHz for CPU search. For cross encoder, we conduct label augmentation by re-ranking text pairs with a batch size of 100. For dense retrieval (Contriever and GTR-XXL) and sparse retrieval, we use Faiss-GPU index and Lucene index from Pyserini (Lin et al., 2021a) with 60 threads, respectively, and search with a batch size of 100. Note that we use a batch size of 25 to encode queries using GTR-XXL due to GPU mem-

Table 10: The latency comparison of relevance label augmentation with batch inference using different teachers on MS MARCO.

| Type | Model | candidates (#) | latency (ms/q) GPU | latency (ms/q) CPU |
|---|---|---|---|---|
| cross-encoder | miniLML6v2 | 1K | 600 | - |
| dense | Contriever | 8.8M | < 1 | - |
| dense | GTR-XXL | 8.8M | 10 | - |
| sparse | uniCOIL | 8.8M | - | 84 |
| sparse | SPLADE++ | 8.8M | - | 144 |
| multi-vec | ColBERTv2 | 8.8M | 55 | - |

Table 11: Detailed effectiveness (Success5) on LoTTE.

| | | sparse 0 SPLADE++ | multi-vec 1 ColBERTv2 | dense C DRAGON-S | dense D DRAGON-Q | dense E DRAGON | dense F DRAGON+ |
|---|---|---|---|---|---|---|---|
| **Search** | writing | 78.7 | 80.1 | 78.8 | 78.2 | 79.2 | 81.5 |
| | recreating | 71.9 | 72.3 | 73.4 | 74.6 | 76.0 | 73.9 |
| | science | 56.6 | 56.7 | 55.3 | 56.9 | 56.9 | 57.9 |
| | technology | 65.9 | 66.1 | 64.9 | 68.8 | 65.4 | 67.6 |
| | lifestyle | 83.7 | 84.7 | 84.9 | 84.7 | 85.6 | 85.9 |
| | pooled | 70.9 | 71.6 | 71.4 | 72.4 | 72.6 | 73.5 |
| **Forum** | writing | 75.2 | 76.3 | 76.2 | 75.2 | 75.6 | 77.5 |
| | recreating | 69.2 | 70.8 | 69.9 | 69.3 | 70.3 | 69.1 |
| | science | 44.9 | 46.1 | 40.1 | 40.1 | 40.7 | 41.4 |
| | technology | 53.1 | 53.6 | 50.5 | 51.2 | 50.5 | 51.4 |
| | lifestyle | 76.9 | 76.9 | 77.0 | 77.4 | 76.9 | 77.7 |
| | pooled | 62.3 | 63.4 | 61.1 | 61.2 | 61.4 | 62.1 |

ory constraint, which is also the main bottleneck for GTR-XXL batch retrieval. For ColBERTv2, we use the improved version of multi-vector retrieval, PLAID (Santhanam et al., 2022a), and search with a batch size of 1, which is the only option.

Table 10 compares the latency cost per query for relevance label augmentation with different neural rankers and demonstrates that leveraging all the retrievers to augment relevance labels are still more efficient than a cross encoder.

### A.8 Detailed evaluation on LoTTE

Table 11 lists DRAGON's effectiveness on five topics without aggregation. Although all the variants of DRAGON show strong effectiveness on LoTTE, we find that DRAGONs perform poorly on the Forum queries about topics of science and technology compared to SPLADE++ and ColBERTv2. Combining science corpus pre-training with DRAGON training strategy is a possible solution.

### A.9 More Related Work

**Knowledge Distillation.** Our work is closely related to the previous work exploring knowledge distillation (KD; Hinton et al., 2015) from ColBERT, cross encoder or their ensemble (Hofstätter et al., 2021; Hofstätter et al., 2020) to improve the effectiveness of DR (Lin et al., 2021b; Qu et al., 2021). However, they only take the advantage of soft labels from KD and use the relevant passages labeled by humans. The recent work (Ren et al., 2021; Zeng et al., 2022) mines more positive samples using cross encoder to further augment the limited relevance labels by humans. Nevertheless, it is challenging for cross encoders to augment relevance labels for queries in scale due to its low efficiency. Chen et al. (2022) first explore label augmentation using singe sparse retrieval model on large-scale queries and demonstrate that a dense retriever can mimic a teacher of a sparse retriever (e.g., BM25). Different from the previous work, we explore label augmentation using multiple supervisions on large-scale augmented queries.

**Curriculum Learning.** Easy-to-hard training strategies (Bengio et al., 2009) have been applied to improve many machine learning tasks, including dense retrieval (Zeng et al., 2022; Lin et al., 2022). The previous work focuses on distilling complex knowledge from cross encoders to a dense retriever with a curriculum training strategy and demonstrates improved effectiveness in supervised retrieval tasks. In our work, we explore to progressively train a dense retriever with the diverse supervisions from dense, sparse and multi-vector retrievers to improve both supervised and zero-shot effectiveness.

**Pre-Training.** There are two popular approaches to pre-training a dense retriever. The first one is contrastive pre-training, aiming to increase the size of training data by creating artificial text pairs (Lee et al., 2019; Chang et al., 2020; Izacard et al., 2021) from a corpus or collecting question–answer pairs (Oguz et al., 2022; Ni et al., 2022) from websites. The second one is masked auto encoding pre-training, where models are trained to recover the corrupted texts (Gao and Callan, 2021; Lu et al., 2021; Xiao et al., 2022; Wu et al., 2022). Our work is similar to contrastive pre-training but instead of creating large-scale training data in an unsupervised or weakly supervised manner, we investigate how to conduct supervised contrastive learning on artificially created text pairs. We demonstrate that

combining masked auto encoding pre-training and our supervised contrastive learning can further improve models' generalization capability.