# OpenReview forum: "How to Train Your  Dragon: Diverse Augmentation Towards Generalizable Dense Retrieval"
_EMNLP/2023/Conference — EMNLP 2023 Findings_

### Official Review · Reviewer_a2hW · 2023-08-03

**Soundness:** 2

**Excitement:**

3: Ambivalent: It has merits (e.g., it reports state-of-the-art results, the idea is nice), but there are key weaknesses (e.g., it describes incremental work), and it can significantly benefit from another round of revision. However, I won't object to accepting it if my co-reviewers champion it.

**Paper Topic And Main Contributions:**

This paper presents a progressive method for data augmentation to enhance query retrieval and obtain a more comprehensive set of retrieval content.






**Questions For The Authors:**

1. The author should provide more technical details, such as code disclosure, to facilitate easy reproduction and evaluation of the proposed method.

2. Additionally, conducting a theoretical analysis of the proposed technology, particularly on how to prevent the propagation of errors generated during data augmentation, would enhance the paper's scientific rigor and reliability.






**Reasons To Accept:**

The experimental results demonstrate the effectiveness of the proposed data augmentation strategy.






**Reasons To Reject:**

The technical details provided in this paper are not sufficient for easy reproduction and lack theoretical analysis, particularly regarding error propagation prevention during data augmentation and discussion on error rates in formulas. Further clarification and additional theoretical analysis are needed to strengthen the paper's scientific rigor and reproducibility.






**Reproducibility:**

2: Would be hard pressed to reproduce the results. The contribution depends on data that are simply not available outside the author's institution or consortium; not enough details are provided.

**Reviewer Confidence:**

2: Willing to defend my evaluation, but it is fairly likely that I missed some details, didn't understand some central points, or can't be sure about the novelty of the work.

---

> ### Author Rebuttal · Authors · 2023-08-25
>
> We thank the reviewer’s helpful suggestions. See below responses to the concerns and questions.
>
> 1. Issue of reproduction:
> We have released all our training and evaluation code, augmented training data and model checkpoints but cannot provide the link for anonymity.
>
> 2. Theoretical analysis of the proposed approach on the prevention of error propagation generated during data augmentation:
> The intuition behind our training strategy is to prevent errors created during data augmentation (DA). As discussed in lines 194--199, we challenge the previous approaches (e.g, contrastive pre-training and GenQ) which assume that the generated queries or segmented sentences are relevant to their original documents. Thus, we propose relevance label augmentation to mine relevant documents for each augmented query. Furthermore, we improve the quality of relevance labeling using multiple teachers. We believe our extensive evaluations on diverse datasets can justify our approach.
>
> Second, while it is challenging to provide a theoretical analysis of a deep neural model without making over-simplifying assumptions, we attempt to provide insights and explanations to our modeling choices and observations whenever possible. For example, we draw inspiration from curriculum learning to explain the result that uniform sampling provides better supervision accuracy than progressive supervision but leads to suboptimal performance (see the detailed analysis in Appendix A.6).

---

### Official Review · Reviewer_1UUD · 2023-08-05

**Typos Grammar Style And Presentation Improvements:** 1. In Table 4, Contriever (8) is list…
**Soundness:** 4

**Excitement:**

4: Strong: This paper deepens the understanding of some phenomenon or lowers the barriers to an existing research direction.

**Paper Topic And Main Contributions:**

This paper discusses two data augmentation approaches that are commonly used in dense retrieval models and proposes a unified framework to leverage the benefits of two approaches to improve dense retrieval. For query argumentation, the proposed method mixes two types of data, namely sentence cropping and synthetic queries. And for relevance label argumentation, author proposed a progressive label method to distill knowledge from multiple and diverse supervisions. The proposed model has smaller size compared with other state-of-the-art models but has comparable performance.

**Questions For The Authors:**

1. In section 5, for cross encoder experiment, what is the performance of CE model in both supervised and zero-shot setting? It seems that CE is added in the last iteration, does this follow the rule of ordering different supervision mentioned in section 3.3 (i.e. augment a supervision at a later stage if it has better generalization capability?)  Why does CE model not provide additional benefit?


**Reasons To Accept:**

1. The proposed method shows state-of-the-art performance on supervised and zero-shot evaluation settings.
2. The proposed data augmentation approaches are less expensive compare with prior work. For example, using sentence cropping in fine-tuning and using retrievers instead of expensive cross-encoder model for labeling.
3.  The paper is mostly well written and easy to follow.

**Reasons To Reject:**

1. Even though using retrievers for labelling are more efficient than cross-encoder models, the proposed approach requires multiple teacher retrievers, which increases the complexity from a different aspect.
2. It is not clear what is the optimal number of supervisions needed for relevance label argumentation.

**Reproducibility:**

4: Could mostly reproduce the results, but there may be some variation because of sample variance or minor variations in their interpretation of the protocol or method.

**Reviewer Confidence:**

4: Quite sure. I tried to check the important points carefully. It's unlikely, though conceivable, that I missed something that should affect my ratings.

---

> ### Author Rebuttal · Authors · 2023-08-25
>
> We thank the reviewer’s insightful suggestions and discussion. See below responses to the concerns and questions.
>
> 1. The proposed approach increases the complexity from a different perspective:
> We agree that conducting auto labeling with multiple retrievers may increase the complexity (e.g., training data creation), although Table 10 shows that using all the retrievers to create training data is still more efficient than using one cross encoder.   In this paper, we want to highlight the importance of diverse supervision signals as opposed to the widely accepted wisdom that single and strong cross-encoder can provide sufficient supervision for DR training. We agree that creating diverse supervision signals with a less complex method is also important and will add this point to our future work.
>
> 2. It is not clear what is the optimal number of supervisions:
> Instead of claiming how many teachers could give the optimal results, our intuition is that as long as we can find a teacher that creates a more diverse signal in our augmentation data, we can still improve our model’s retrieval effectiveness. See Table 9 in Appendix A.6 on how we measure the diversity of our progressively augmented data.
>
> 3. Performance of CE model and do you follow the rule of supervision ordering:
> We use the cross-encoder ``cross-encoder/ms-marco-MiniLM-L-12-v2`` from Huggingface mentioned in Appendix A.2; its ranking effectiveness (using DRAGON as the first-stage retriever) is above 0.4 in MRR@10 on MS MARCO development queries. Although we did not measure its zero-shot effectiveness on BEIR, the results from [1] shows that ``cross-encoder/ms-marco-MiniLM-L-6-v2``, the smaller version of our cross-encoder renaker, reaches over 0.49 on BEIR 18 datasets. Thus, we assume that our CE model has better zero-shot effectiveness than the other teachers and the experiments in section 5 do follow the rule of supervision ordering we proposed.
>
> 4. Why does the CE model not provide additional benefit:
> We hypothesize that the CE supervision cannot create a more diverse signal since the training data is created by re-ranking DRAGON top-1000 results, which is still very similar to the original DRAGON ranked list.
>
> [1] Guilherme Moraes Rosa, Luiz Bonifacio, Vitor Jeronymo, Hugo Abonizio, Marzieh Fadaee, Roberto Lotufo, and Rodrigo Nogueira. No parameter left behind: How distillation and model size affect zero-shot retrieval. arXiv preprint arXiv:2206.02873, 2022.

---

### Official Review · Reviewer_CLak · 2023-08-09

**Soundness:** 3

**Excitement:**

4: Strong: This paper deepens the understanding of some phenomenon or lowers the barriers to an existing research direction.

**Paper Topic And Main Contributions:**

The paper presents a data augmentation-based pipeline, which allows the model to achieve Retrieval State-of-the-Art (SOTA) performance while maintaining the parameter count of BERT-base.

**Questions For The Authors:**

Is there a possibility that DRGAON experiences overfitting or even model collapse due to augmented data? Could you delve deeper into this aspect?

**Reasons To Accept:**

The motivation is well-founded. The explored content in the article demonstrates an interesting phenomenon: that we can further enhance performance while keeping or even reducing the model's size. This may inspire in the era of large models.

**Reasons To Reject:**

The discussion about the negative impact of augmented data in the article is not sufficient. The presence of a significant amount of augmented data might impose limitations on the model's performance.

**Reproducibility:**

4: Could mostly reproduce the results, but there may be some variation because of sample variance or minor variations in their interpretation of the protocol or method.

**Reviewer Confidence:**

3: Pretty sure, but there's a chance I missed something. Although I have a good feel for this area in general, I did not carefully check the paper's details, e.g., the math, experimental design, or novelty.

---

> ### Author Rebuttal · Authors · 2023-08-25
>
> We thank the reviewer’s insightful discussion. See below responses to the concerns and questions.
>
> 1. Potential  negative impact of augmented data / Possibility of DRAGON overfitting due to augmented data:
> It is indeed the central focus of our paper to train a more generalizable retriever and avoid overfitting. Through a systematic study of various data augmentation approaches, we believe that our “diverse” augmentation strategy yields a more generalizable retriever compared to existing works. This is evidenced by the extensive zero-shot evaluations we conducted on benchmarks like BEIR and LoTTE. This experimental setting makes sure our model does not see the evaluation data (even corpus) during training. The fact that DRAGON achieves a state-of-the-art performance on these datasets shows that the diversity of our query and relevance label augmentation ensures the effectiveness of DRAGON across a wide range of domains and datasets.

---

### Official Review · Reviewer_VbrU · 2023-08-12

**Soundness:** 3

**Excitement:**

4: Strong: This paper deepens the understanding of some phenomenon or lowers the barriers to an existing research direction.

**Paper Topic And Main Contributions:**

This paper proposes a Data Augmentation framework for Dense Retrieval models (DRAGON), allowing BERT-size dense retrieval models to achieve state-of-the-art results. Dragon consists of both query augmentation (sentence cropping and psuedo-query generation) and relevance label augmentation(multiple retrievers for different relevancies). Authors follow standard settings for training ad evaluation of their model using MSMarco training corpus as training and DA source and MSMarco Dev set for supervised evaluation and BEIR benchmark for zero-shot evaluation. The paper explores three different training supervision (Fused, Uniform, Progressive) to utilize different rank lists obtained from diverse relevancy labels.

**Reasons To Accept:**

The paper is well-written and well-structured. It investigates the important problem of the capacity of BERT-size models for dense retrieval in the presence of multiple approaches for data augmentation. The paper explores various approaches for training using multiple sources of label relevancy. The experimentation shows the effect of each query augmentation approach with each of the training supervision approaches.

**Reasons To Reject:**

Statistical significance is not reported for improvement which makes it challenging to interpret the results.
The authors mentioned they compare Dragons with the dense retrieval models with the backbone of bert-base-uncased trained with advanced techniques (see line 474), however, they claim that Dragons achieve state-of-the-art results without increasing the model capacity. This statement prompts a comparison of Dragons to other methods that use larger models.

**Reproducibility:**

4: Could mostly reproduce the results, but there may be some variation because of sample variance or minor variations in their interpretation of the protocol or method.

**Reviewer Confidence:**

4: Quite sure. I tried to check the important points carefully. It's unlikely, though conceivable, that I missed something that should affect my ratings.

---

> ### Author Rebuttal · Authors · 2023-08-25
>
> We thank the reviewer’s insightful discussion.
>
> 1. Statistical significance for improvement is not reported:
> While we agree that conducting statistical significance tests can provide a more solid conclusion, we believe that our experiments sufficiently demonstrates DRAGON’s clear win over other baseline models. For example, in Table 11 (in Appendix), we report the detailed numbers of 18 BEIR datasets and DRAGON+ shows the best or second best result over 12 BEIR datasets. More importantly, DRAGON+ is the only DR model reaching state-of-the-art performance in both supervised (MS MARCO0) and zero-shot (BEIR, LoTTE) evaluations. Finally, as we mentioned in our paper, some compared models using ``non-standard’’ MS MARCO corpus augmented with titles or assuming BEIR corpus are available while training or training one expert model for each dataset. Those approaches have unfair advantages in either MS MARCO or BEIR dataset. Thus, it may not be easy to claim a significant win over those models.
>
> 2. Comparison to other larger models:
> We agree with this point, and we’ve reported GTR-XXL’s (4.8B) performance in the main comparison. Other larger bi-encoder models worth mentioning are CPT_XL (175B), E5_large (335M) and COCO-DR large (335M). As they adopt different subsets of BEIR, we provide the averaged numbers for their common BEIR datasets (see CPT Sub in [1]) as reference. The corresponding average numbers for CPT_XL, E5_large and COCO-DR large are 52.7, 53.0 and 54.1 compared to 53.0 for DRAGON+. However, as far as we know, most of the larger generalizable dual encoders cannot be fairly compared with DRAGON except for GTR. For example, COCO-DR large [1], as we highlighted, is pre-trained on all the corpus in BEIR datasets. In addition, E5 large [2] is fine-tuned on both MS MARCO and NQ (NQ is included in BEIR evaluation and multiple datasets in BEIR use wikipedia as corpus). We will explicitly mention this in the paper if accepted.
>
>
> [1] Yue Yu, Chenyan Xiong, Si Sun, Chao Zhang, and Arnold Overwijk. 2022. Coco-dr: Combating distribution shifts in zero-shot dense retrieval with contrastive and distributionally robust learning. In Proc. EMNLP.
> [2] Liang Wang, Nan Yang, Xiaolong Huang, Binxing Jiao, Linjun Yang, Daxin Jiang, Rangan Majumder, and Furu Wei. 2022b. Text embeddings by weakly-supervised contrastive pre-training. ArXiv, abs/2212.03533.

---

### Meta-Review · Area_Chair_8rjv · 2023-09-27

**Recommendation:** 4

**Metareview:**

This paper proposes a new DA approach with diverse queries and sources of supervision to progressively train a generalizable dense retrieval. Although there are varying opinions among the reviewers, the majority have given positive scores. Reviewers have raised concerns on comparison with larger model or potential model collapse when enhancing  with negative samples, or model complexity. The rebuttal has addressed these concerns. These suggestions by reviewers have contributed to strengthening this paper. Reviewers have high excitement score to this paper.

---

### Decision · Program_Chairs · 2023-10-07

**Decision:**

Accept-Findings

**Comment:**

This paper proposes a new DA approach with diverse queries and sources of supervision to progressively train a generalizable dense retrieval. Although there are varying opinions among the reviewers, the majority have given positive scores. Reviewers have raised concerns on comparison with larger model or potential model collapse when enhancing  with negative samples, or model complexity. The rebuttal has addressed these concerns. These suggestions by reviewers have contributed to strengthening this paper. Reviewers have high excitement score to this paper.